# *KRAS* G12D Mutation Subtype in Pancreatic Ductal Adenocarcinoma: Does It Influence Prognosis or Stage of Disease at Presentation?

**DOI:** 10.3390/cells11193175

**Published:** 2022-10-10

**Authors:** Henry Shen, Joanne Lundy, Andrew H. Strickland, Marion Harris, Michael Swan, Christopher Desmond, Brendan J. Jenkins, Daniel Croagh

**Affiliations:** 1Department of Surgery, Faculty of Medicine, Nursing and Health Sciences, Monash University, Clayton, VIC 3800, Australia; 2Centre for Innate Immunity and Infectious Diseases, Hudson Institute of Medical Research, Clayton, VIC 3168, Australia; 3Department of Molecular and Translational Science, Faculty of Medicine, Nursing and Health Sciences, Monash University, Clayton, VIC 3800, Australia; 4Department of Oncology, Faculty of Medicine, Nursing and Health Sciences, School of Clinical Sciences, Monash University, Clayton, VIC 3800, Australia; 5Department of Gastroenterology, Monash Medical Centre, Monash Health, Clayton, VIC 3168, Australia

**Keywords:** carcinoma, pancreatic ductal, proto-oncogene proteins p21(ras), prognosis, survival, point mutation

## Abstract

**Background**: *KRAS* G12D mutation subtype is present in over 40% of pancreatic ductal adenocarcinoma (PDAC), one of the leading global causes of cancer death. This retrospective cohort study aims to investigate whether detection of the *KRAS* G12D mutation subtype in PDAC patients is a determinant of prognosis across all stages of disease. **Methods**: We reviewed the medical records of 231 patients presenting with PDAC at a large tertiary hospital, and compared survival using the Kaplan Meier, log-rank test and Cox proportional hazards regression model. **Results**: *KRAS* G12D mutation subtype was not significantly associated with poorer survival compared across the whole population of PDAC patients (*p* = 0.107; HR 1.293 95% CI (0.946–1.767)). However, *KRAS* G12D patients who were resectable had a shorter median survival time of 356 days compared to all other genotypes (median survival 810 days) (*p* = 0.019; HR 1.991 95% CI (1.121–3.537)). **Conclusions**: *KRAS* G12D patients who were resectable at diagnosis had shorter survival compared to all other PDAC patients. These data suggest that *KRAS* G12D may be a clinically useful prognostic biomarker of PDAC.

## 1. Introduction

Globally, pancreatic cancer was the 12th most newly diagnosed cancer in 2020, with 495,773 new cases, and the 7th leading cause of cancer death in 2020 with 466,003 deaths [1,2]. Currently, the only cure for pancreatic cancer is surgery [3], however, only 15–20% of patients have resectable tumours at diagnosis [3]. Furthermore, up to 80% of patients have tumour recurrence following resection [4]. There are few clinically available biomarkers that can effectively guide prognosis and treatment in pancreatic ductal adenocarcinoma (PDAC). Measuring serum levels of cancer antigen 19-9 (CA19-9) can be useful to confirm diagnosis as well as monitor disease recurrence. However, it has a relatively low predictive value and is of limited use in selecting patients for resection [5].

Endoscopic ultrasound guided fine needle aspiration (EUS-FNA) remains the gold standard for the diagnosis of localised PDAC, with a sensitivity of around 85% and specificity of near 100% [6]. Furthermore, it can be used to provide tissue for cytological and even histological diagnosis and can also be utilised for further applications such as *KRAS* detection [7]. However, it is not currently used to provide prognostic information to guide selection of patients for surgical resection.

In recent years, the molecular landscape of pancreatic cancer has become well defined. The four most commonly mutated genes in pancreatic cancer are *KRAS*, *CDKN2A*, *TP53*, and *SMAD4*. Mutations in these genes have been associated with oncogenic cellular processes and with prognosis in pancreatic cancer [8]. The *KRAS* gene encodes a guanosine triphosphatase (GTPase) protein that controls cellular processes by linking membrane growth factor receptors to intracellular signalling pathways and transcription factors. The *KRAS* gene is mutated in over 90% of PDAC cases [9].

The *KRAS* mutation is usually a point mutation which frequently occurs on codon 12 of exon 2, affecting the first or second nucleotide, but can also affect other codons and exons such as codon 13 and 61. The normal or wild-type sequence is GGT encoding glycine, and it is present in 8–12% of PDAC [10]. The most common mutation subtype is G12D, present in 40% of patients, whereby a GAT sequence is coded, producing aspartic acid [10]. G12V subtype refers to a GTT replacement sequence producing valine, and G12R refers to a CGT replacement sequence producing arginine, and these are present in 33% and 15% of PDAC cases, respectively [10].

Detection of *KRAS* does not currently have a role in screening or diagnosis of pancreatic cancer due to difficulty standardizing sampling, transportation, extraction and detection. However, many studies have demonstrated cytopathology with *KRAS* on EUS-FNA materials has a higher sensitivity, specificity and accuracy of PDAC diagnosis compared to cytopathology alone [10]. Furthermore, liquid biopsies which detect circulating tumour cells and cell-free circulating tumour DNA (ctDNA) have been heavily investigated diagnostically and prognostically, with varying results [10,11].

Many studies have investigated the association between *KRAS* mutation and PDAC patient survival, predominantly in patients with early-stage, resectable disease. Most studies found that *KRAS* mutation was significantly associated with shorter overall survival, although a small number of studies found no statistically significant correlation [12,13,14,15]. The association between *KRAS* mutation and survival in PDAC patients with advanced unresectable disease has also been assessed. EUS-FNA was the main source of tissue for *KRAS* detection in this patient cohort [16,17,18]. Ogura et al. analysed 242 patients with unresectable pancreatic cancer and found that those in the *KRAS* mutation group had a significantly shorter survival compared to the wild-type group [16]. Conversely, a recent study by on 219 patients with advanced PDAC found that there was no significant difference in survival between the mutant *KRAS* and *KRAS* wild-type groups [18]. However, overall, as demonstrated in a recent meta-analysis, patients with *KRAS* mutation had poorer overall survival [19].

When considering the different mutant *KRAS* subtypes, G12D is both the most frequently occurring *KRAS* mutation subtype (36.9–67%) and also the subtype that is most commonly reported as being associated with poorer survival [12,13,14,15,16,18,20,21]. For example, Qian et al. found that *KRAS* G12D subtype patients had an overall median survival of 15.3 months compared to 24.8 months in patients without the *KRAS* G12D subtype [15] (*n* = 356). However, the *KRAS* G12A and G12R subtypes have also been reported as being associated with a reduction in patient survival [12,16], while other investigators have been unable to demonstrate any correlation between a particular *KRAS* subtype and survival [17].

Given the heterogeneity of these results, we sought to assess the correlation between the *KRAS* G12D mutation subtype and survival in pancreatic cancer patients across all clinical stages using tissue biopsies obtained predominantly via EUS-FNA.

## 2. Methods

### 2.1. Ethics Statement

This retrospective cohort study was approved by the Monash Health Human Research Ethics Committee (Monash Health HREC Ref: 17387L). The cohort was comprised of patients who had previously had *KRAS* mutations analysis performed on EUS-FNA or resection specimens. The patients had either provided tissue as part of the Victorian Pancreatic Cancer Biobank and various sub studies associated with this (Monash Health HREC Ref: 15450A) or were being screened for enrolment in a prospective cohort study examining the use of panitumumab as second or subsequent line therapy in advanced *KRAS* wild-type pancreatic cancer (Monash Health HREC Ref: 16584A) [22]. All patient information was stored in a confidential and de-identified manner.

### 2.2. Data and Sample Collection

We included all consenting patients from 2012 to 2020 with *KRAS* mutation data. Demographic data, date and method of diagnosis, treatment details and survival were reviewed. Clinical stage was established via review of multi-disciplinary team meeting documentation. In general, clinical staging was determined on anatomical criteria using the NCCN Pancreatic Adenocarcinoma guidelines (2016) [23,24]. These guidelines specify criteria such as degree of tumour contact or invasion on the superior mesenteric artery, coeliac artery, or common hepatic artery, to distinguish between resectable, borderline resectable, locally advanced or metastatic pancreatic disease.

*KRAS* mutation analysis was performed either on EUS FNA specimens or from tissue taken from pancreatic resections (either a snap frozen core biopsy of the tumour taken immediately after resection or from archival formalin fixed paraffin blocks later retrieved from the pathology department). For patients undergoing EUS-FNA an additional biopsy was snap frozen and stored at −80°C or in liquid nitrogen in the Victorian Pancreatic Cancer Biobank (Monash Health HREC Ref: 15450A) or the Monash Surgical Oncology Biobank (Monash Health HREC Ref: 13058A).

DNA was extracted from patient biopsy samples via homogenization in a Buffer RLT plus AllPrep DNA/RNA Universal Kit (Qiagen, Hilden, Germany) as per the manufacturer’s protocol. The isolation of gDNA from FFPE tissue was performed on 5 × 10 micron-thick sections using the ReliaPrep FFPE gDNA Miniprep System (Promega, Madison, WI, USA). The quality and quantity of gDNA were determined on a NanoDrop Spectrophotometer (Thermo Fisher Scientific, Waltham, MA, USA) and Qubit Fluorometer (ThermoScientific), and TapeStation (Agilent, St. Clara, CA, USA). gDNA (25–50 ng) was subjected to the *KRAS* XL StripAssay^®^ (ViennaLab Diagnostics GmbH, Wien, Austria). Mutations were objectively scored using StripAssay Evaluator software. For a small number of patients, *KRAS* mutation was determined using the TruSight Oncology 500 (TSO-500) gene panel (Illumina, San Diego, CA, USA).

### 2.3. Statistical Analysis

Survival was measured from date of first tissue diagnosis (either EUS-FNA or surgical resection, whichever came first) until death or censoring date defined as the last known follow-up. Treatment groups were based on if patients received resection with curative intent, chemotherapy and/or radiotherapy without surgery or supportive care only. In patients who received surgery, we further recorded lymph node status, tumour location, resection margin clearance and whether or not adjuvant or neoadjuvant therapy was administered.

A chi-squared test, chi-squared test with Yates’ continuity correction, and Fisher’s exact test were used to assess independent categorical variables. A Mann—Whitney U test was used to compare differences in age. Kaplan-Meier curves were generated for median survival within individual groups, and these were compared using the log-rank test for univariate analysis. *KRAS* mutant was compared with *KRAS* wild-type, and the G12D subtype was compared with non-G12D patients, which consists of all other mutant *KRAS* subtypes and *KRAS* wild-types combined. Multivariate analysis only included *KRAS* mutation as a whole, and also the G12D, G12V and G12R subtypes individually, and used covariates associated with survival on univariate analysis (defined by *p* < 0.1) and was conducted using the Cox proportional hazards regression model. The sample method (FNA vs. resection) was excluded from multivariate analysis as this was not considered clinically relevant to patient survival.

Statistical significance was defined at *p* < 0.05. All statistical analyses were performed using GraphPad Prism Version 9.1.1 (GraphPad Software, San Diego, CA, USA) and IBM SPSS Statistics 27 (IBM, Armonk, NY, USA).

## 3. Results

### 3.1. Patient Characteristics at Baseline

Baseline characteristics are shown in Table 1. A total of 231 patients had *KRAS* mutation status available for analysis (109 males, 122 females, IQR = 14.02). Of these, 202 patients had a *KRAS* mutation (87.4%) and 29 patients were *KRAS* wild-type (12.6%). Furthermore, 194 patients had a *KRAS* mutation detected through EUS-FNA (84.0%) with a *KRAS* mutation rate of 87.1% (169/194), whilst 37 patients had a *KRAS* mutation detected through assessment of resection specimens (16.0%) with a mutation rate of 89.2% (33/37). There was no difference in the KRAS detection rate depending on the method of tissue acquisition, either EUS-FNA or surgical resection (*p* = 0.9374).

The 3 most frequently occurring *KRAS* mutation subtypes were G12D (93 patients, 40.3%), G12V (64 patients, 27.7%) and G12R (24 patients, 10.4%). (Table 2) There were 138 patients in total who were not *KRAS* G12D, including *KRAS* wild-type patients (59.5%).

There was no significant correlation between *KRAS* mutation and clinical stage at diagnosis (*p* = 0.1627; df = 3). Further, the G12D mutation subtype was also not significantly associated with clinical stage at diagnosis (*p* = 0.500, df = 3) (Table 1).

For patients who underwent resection, lymph node status (positive or negative), tumour location within pancreas (head or distal), neoadjuvant therapy (yes or no), resection margin (clear or positive) and adjuvant therapy (yes or no) information was present for 50, 53, 51, 53 and 51 patients, respectively, due to loss of follow-up (see Table 3). In patients who received surgery, there was no correlation between *KRAS* mutation status or G12D subtype and surgical parameters such as lymph node status (*p* = 0.8719, 95% CI 0.6179–2.760) and tumour location (*p* = 0.9274, RR 1.170 95% CI 0.6013–2.694) (Table 3).

### 3.2. Survival Analysis for the Entire Cohort

Multivariate analysis showed that only NCCN clinical stage and treatment modality were significantly associated with overall survival (Table 4). Across the whole population, patients with a *KRAS* mutation did not have significantly worse survival compared to *KRAS* wild-type patients (296 vs. 420 days, *p* = 0.843; HR 1.050 95% CI (0.646–1.709)). The G12D mutation subtype was also not associated with poorer survival compared to non-G12D PDAC patients (which included both other forms of mutant *KRAS* and *KRAS* wild-type) (*p* = 0.107; HR 1.293 95% CI (0.946–1.767)). Similarly, neither G12V nor G12R were associated with changes in patient survival.

In patients who received surgery, lymph node positivity and tumour location were associated with survival on univariate analysis but not on multivariate analysis. Neoadjuvant therapy, resection margins and adjuvant therapy were not associated with survival in this cohort (Table 5).

### 3.3. Survival Analysis by NCCN Stage and Treatment Modality and KRAS Mutation Status

There was no significant difference in overall survival between *KRAS* mutant and wild-type patients in any clinical stage of disease, however, in the resectable group, G12D patients had a significantly shorter median survival time of 356 days compared to non-G12D PDAC patients (which included both other forms of mutant *KRAS* and *KRAS* wild-type) (median survival 810 days) on both univariate analysis (*p* = 0.0168; HR 2.053 95% CI (1.139–3.702)) and multivariate analysis (*p* = 0.019; HR 1.991 95% CI (1.121–3.537)) (Table 6 and Figure 1). There were no associations between survival and G12D subtype in any of the other stages of disease, namely, borderline resectable, locally advanced and metastatic.

Similarly, with respect to the treatment modality, there was also no statistically significant difference in overall survival between *KRAS* mutant and wild-type patients in patients in any of the treatment groups: surgery, chemotherapy/radiotherapy and best supportive care. However, G12D patients who underwent surgery had significantly shorter median survival of 462 days compared to all non-G12D patients (median survival 1084 days) on univariate analysis (0.0221; HR 2.000 95% CI (1.079–3.707)) although not on multivariate analysis (0.254; HR 1.471 95% CI (0.758–2.855)) (See Figure 1). Within the chemotherapy/radiotherapy group and also within the supportive care group, there was no association between *KRAS* G12D subtype and survival.

These data suggest that the *KRAS* G12D mutation subtype is associated with poorer survival within NCCN resectable patients. Due to the low numbers of G12V and G12R patients within each clinical stage and treatment modality cohort, no attempt was made to assess for correlations between survival and the presence of these subtypes.

## 4. Discussion

In this retrospective cohort study examining the correlation between the presence of a *KRAS* G12D mutation subtype and prognosis in PDAC across all stages of disease, we found that the presence of a *KRAS* G12D mutation was associated with reduced survival in those with resectable disease. There was also a suggestion (on univariate analysis only) of reduced survival in patients with a *KRAS* G12D mutation who underwent resection, however, after adjusting for potential confounders, this was no longer statistically significant (on multivariate analysis). A larger sample size may help clarify this further. There was no association between *KRAS* G12D mutation and the clinical stage of disease at presentation. Further, we could not demonstrate a statistically significant association between the presence of a *KRAS* G12D mutation subtype and survival in other clinical stages of disease or with other treatment modalities.

Our study is unique in a number of ways. Firstly, although we predominantly used EUS to assess *KRAS* mutation status, we also included analysis of resection specimens (in patients who had not undergone EUS-FNA) allowing us to assess the association between *KRAS* mutation and subtype across all stages of disease and in all treatment groups. Furthermore, we found that 87.4% of patients had a *KRAS* mutation, which reflects the approximately 90% frequency reported in current literature [9]. In addition, G12D is typically cited as occurring in 40% of PDAC patients, G12V usually occurs in 33% of patients, and G12R typically occurs in 15% of patients [10]. Our cohort had 40.3% G12D, 27.7% G12V and 10.4% G12R. As such, our cohort is a representative sample of a PDAC population from a *KRAS* mutation frequency perspective. Previous studies which have used EUS-FNA to assess *KRAS* mutation status have often reported significantly lower rates of *KRAS* mutation detection suggesting significant sampling error [12,13,14]. The fact that we utilized an additional snap frozen biopsy for molecular analysis probably accounts for this difference. If EUS-FNA is to be used to guide treatment decisions based on molecular analyses, it is important that the information provided is indeed reliable and reflective of the tumour.

Secondly, our study represents the first time that the prevalence of *KRAS* mutations has been simultaneously analysed across all clinical stages of disease allowing us to assess any potential associations between *KRAS* mutations and both stage at presentation and survival. Most previous studies assessed either only resected patients [25] or advanced unresectable disease [19]. Furthermore, although some previous studies which focused on resected patients considered TNM as an independent predictor of survival [19,25], we chose to use the NCCN staging system [26,27]. The rationale for this is that the TNM staging system requires accurate assessment of lymph nodes status and this is only available for the 20% of patients who undergo resection, whereas the NCCN staging system can be applied to the whole population. We hypothesized that given that treatment options such as surgery or chemotherapy remain the most important prognosticators in PDAC, any association between *KRAS* mutation and survival may be mediated by differences in the prevalence of *KRAS* mutations and/or subtypes in the various NCCN clinical stages at diagnosis. However, as demonstrated above, there was in fact no association between *KRAS* mutation and/or subtype and clinical stage at presentation. Moreover, there was no association between G12D mutation and the presence of lymph node metastases within the resected cohort, further suggesting that the deleterious effect of G12D is not mediated through its impact on stage at diagnosis, even in those with resectable disease.

The G12D subtype has previously been shown to be associated with poorer prognosis, however, the literature is somewhat conflicting [12,13,14,15,17,19,20]. The strongest evidence for G12D being a marker of poor prognosis comes from studies of patients with operable disease, whereas the evidence for advanced disease is less clear cut. Our data support this. Since G12D is present in 40% of PDAC patients [10], this finding has potentially important clinical implications. For patients who are considered anatomically resectable, but who are marginal surgical candidates due to age and comorbidities, *KRAS* mutation assessment could be useful to guide treatment decisions. The fact that *KRAS* mutation subtype can be reliably assessed via EUS-FNA enables this information to be available prior to considering surgery.

The reason for exploring *KRAS* mutation status as a prognostic biomarker rather than any other potentially useful biomarkers, such as Ki67, is that these patients were being screened for potential inclusion in a clinical study of *EGFR* inhibition in *KRAS* wild-type pancreatic cancer [22]. The recent introduction of targeted therapies directed at specific *KRAS* subtypes highlights the fact that it will become increasingly important to be able assess prognostic and predictive biomarkers using EUS-FNA biospecimens. For example, targeting G12C has recently demonstrated significant anti-tumour activity in clinical trials, some of which included PDAC patients [28]. Sotorasib, a G12C inhibitor, has been approved for clinical use in non-small-cell lung cancer, after promising phase I and II trials [28,29]. Another approach includes siG12D-LODER™, an siRNA which targets *KRAS* loaded into a biodegradable polymetric matrix [30]. Early phase clinical trials in PDAC have reported it is well tolerated, with promising signs of efficacy [31]. Engineered inhibitory exosomes have proven to be effective in PDAC preclinical models [32]. All these approaches rely on accurate assessment of the subtype status prior to surgery. We have convincingly demonstrated that EUS-FNA can be used to assess *KRAS* subtype status in patients with PDAC.

There are a number of limitations in our study. These include that it is from a single center and has a relatively small sample size for survival analysis. The latter may have contributed to the failure to demonstrate an association between the presence of a *KRAS* mutation and decreased survival (compared to wild-type disease), despite the positive findings of previous papers. In addition, it was difficult to solely compare G12D subtype survival to *KRAS* wild-type. The relatively small sample size guided our choice to focus on *KRAS* G12D as the most prevalent subtype and to compare this subtype to all other subtypes combined, including G12R, G12V and wild-type. A comparison between each of the various subtypes individually was not possible with this sample size and may have led to the problem of multiple testing. However, comparing G12D to non G12D includes all patients rather than just wild-type patients and is used in existing studies [11,19]. Further, we know that adjuvant therapy, lymph node status and surgical margins are associated with survival outcomes [33,34,35], and the fact that none of these factors were associated with survival on multivariate analysis further demonstrates the lack of power in our study. Finally, the exact details of the chemotherapy received were not reported as many patients had been referred from other centers for diagnostic biopsy and returned to the referring center for chemotherapy. Furthermore, the chemotherapy regimens used were too heterogenous and, given the sample size, were unlikely to have provided statistically meaningful results when examined individually. Time to recurrence was also not reported as it is difficult to collect in retrospective studies, especially as many patients returned to their referring centers for ongoing treatment.

A potential perceived limitation of our study was that only 37 out of the 63 resected specimens had *KRAS* mutation analysis performed on the specimens themselves, with the rest having *KRAS* analysis conducted on EUS-FNA samples prior to surgery. It could be argued that due to the small volume of material obtained, EUS-FNA evaluation may not be able to reflect tumour heterogeneity as well as resection specimens. However, the presence of more than one *KRAS* mutation subtype within the same tumour has almost never been reported even in resection specimens, demonstrating that tumour heterogeneity is not a relevant consideration with respect to *KRAS* mutation subtype. This is probably because an activating *KRAS* mutation is known to occur early in the dysplasia-carcinoma pathway and is therefore likely to be present in virtually every malignant cell.

Given the conflicting results in the literature, further large cohorts may be required to definitively establish the nature of any subtle correlation between NCCN stage and/or prognosis and *KRAS* mutation subtype, particularly G12D, in PDAC. A meta-analysis assessing the association between specific *KRAS* subtypes such as G12D and prognosis in different stages of disease is also warranted given the potential importance of this prognostic information.

## Figures and Tables

**Figure 1 cells-11-03175-f001:**
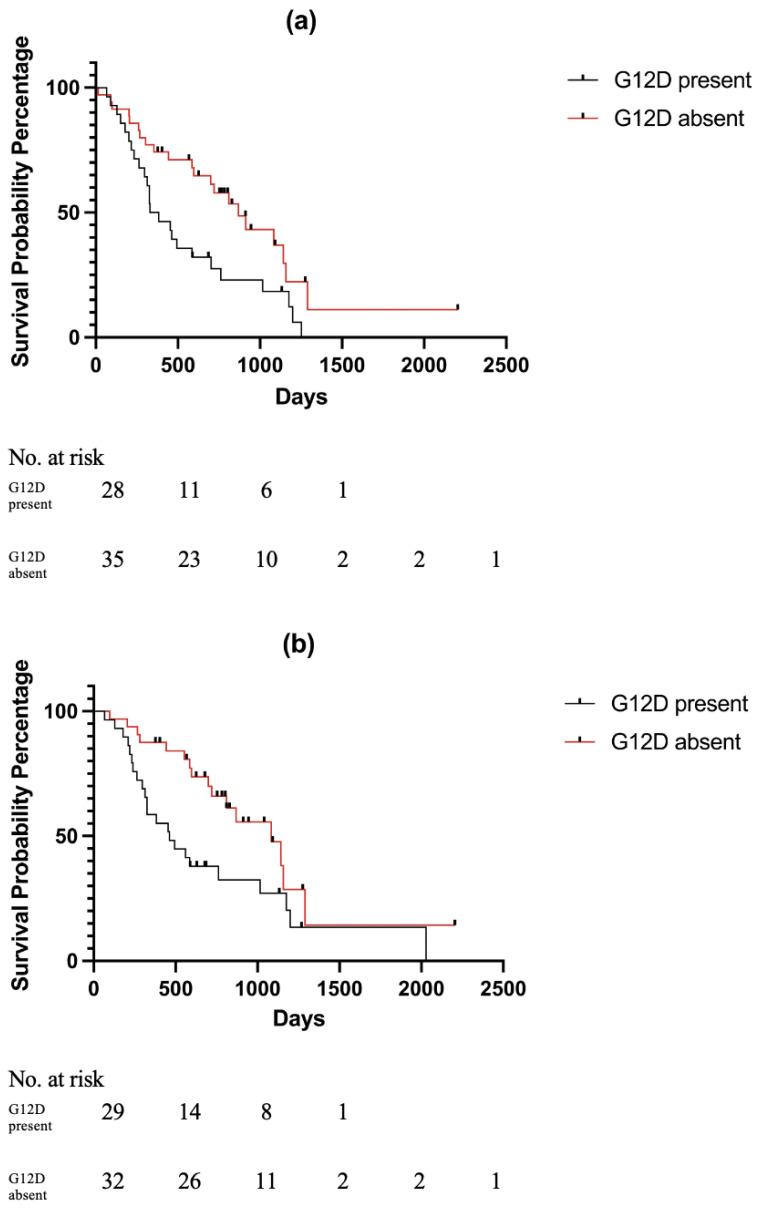
Kaplan Meier curves and respective numbers at risk. G12D subtype associated with poorer survival in patients NCCN staged as resectable. (**a**) G12D subtype associated with poorer survival in patients NCCN staged as operable. (**b**) G12D subtype associated with poorer survival in patients who received surgery. G12D absent includes both other *KRAS* subtypes and *KRAS* wild-type.

**Table 1 cells-11-03175-t001:** Patient characteristics.

			*KRAS*			G12D		
Characteristics		Total	Mutant	WT	*p* Value	Yes	No	*p* Value
Total		231	202	29		93	138	
Sex	Male	109	91	18	0.1290	39	70	0.2388
	Female	122	111	11		54	68	
Median age		70.91	70.91	70.63	0.2562	71.11	70.33	0.2533
Detection method	EUS-FNA	194	169	25	0.9374	75	119	0.3409
	Resection specimen	37	33	4		18	19	
NCCN stage	Resectable	63	55	8	0.1627	28	35	0.500
	Borderline Resectable	23	23	0		9	14	
	Locally Advanced	67	55	12		22	45	
	Metastatic	78	69	9		34	44	
Treatment	Surgery	63	52	9	0.7604	29	32	0.3471
	Chemotherapy and/or Radiotherapy	125	111	14		46	79	
	Supportive Care	43	37	6		16	27	

Table 2: Patient characteristics of whole cohort. Number from each group displayed. Correlation assessed using chi-square test with Yates’ continuity correction and Fisher’s exact test. WT, wild-type; EUS-FNA, endoscopic ultrasound fine needle aspirate; NCCN, National Comprehensive Cancer Network.

**Table 2 cells-11-03175-t002:** *KRAS* mutation subtype frequency.

Subtype	Number of Patients	Frequency
G12A	2	0.9%
G12C	4	1.7%
G12D	93	40.3%
G12R	24	10.4%
G12V	64	27.7%
G13D	2	0.9%
Q61H	11	4.8%
Q61R	2	0.9%
WT	29	12.6%

Table 2: *KRAS* mutation subtype frequency. WT, wild-type.

**Table 3 cells-11-03175-t003:** Surgical characteristics in patients who received surgery.

			*KRAS*			G12D		
Characteristics		Total	Mutant	WT	*p* Value	Yes	No	*p* Value
Lymph node status	Negative	13	12	1	0.2149	5	8	0.7652
	Positive	37	33	4		18	19	
Tumour location	Head	40	33	7	0.6643	18	22	0.8719
	Distal	13	12	1		5	8	
Surgical Margins	Clear	27	24	3	>0.999	12	15	0.9096
	Positive	24	20	4		11	13	
Neoadjuvant therapy	No	10	36	7	0.7030	18	25	0.8990
	Yes	43	9	1		5	5	
Adjuvant therapy	No	12	9	3	0.3720	4	8	0.5095
	Yes	39	34	5		19	20	

Table 3: Surgical characteristics in patients who received surgery. Lymph node status, tumour location within pancreas, neoadjuvant therapy, resection margin and adjuvant therapy information was present for 50, 53, 51, 53 and 51 patients, respectively, due to loss of follow-up. Correlation assessed using chi-square test with Yates’ continuity correction and Fisher’s exact test. WT, wild-type; endoscopic ultrasound fine needle aspirate; NCCN, National Comprehensive Cancer Network.

**Table 4 cells-11-03175-t004:** Independent prognostic factors.

		Median Survival (Days)	Number of Patients	Univariate Analysis (*p*-Value; HR 95% CI)	Multivariate Analysis (*p*-Value; HR 95% CI)
*KRAS*	WT	420.0	202	Reference	Reference
	Mutant	296.0	29	0.2971;	0.843;
				1.257	1.050
				(0.8416–1.877)	(0.646–1.709)
G12D	No	296.0	93	Reference	Reference
	Yes	313.0	138	0.2151;	0.107;
				1.205	1.293
				(0.8934–1.606)	(0.946–1.767)
G12V	No	301.0	167	Reference	Reference
	Yes	295.0	64	0.5200;	0.618;
				0.8992	0.918
				(0.6548–1.235)	(0.656–1.285)
G12R	No	307.0	207	Reference	Reference
	Yes	268.0	24	0.3059;	0.641;
				1.260	1.119
				(0.7741–2.049)	(0.697–1.797)
Sex	Male	301.0	109	Reference	-
	Female	295.0	122	0.4866;	-
				1.106	
				(0.8321–1.470)	
Age	≤70.91	397.0	116	Reference	Reference
	>70.91	228.0	115	0.0135;	0.052
				1.425	0.732;
				(1.070–1.899)	(0.534–1.003)
Clinical stage	Resectable	596.0	63	Reference	Reference
	Borderline Resectable	296.0	23	0.2633;	0.986;
				1.365	1.005
				(0.7488–2.487)	(0.558–1.812)
	Locally Advanced	316.0	67	<0.0001;	0.833;
				2.068	1.061
				(1.383–3.093)	(0.614–1.834)
	Metastatic	193.0	78	<0.0001;	0.047;
				3.277	1.775
				(2.194–4.894)	(1.007–3.128)
Treatment	Surgery	762.0	63	Reference	Reference
	Chemo/Radio	292.0	125	<0.0001;	0.002;
				2.860	2.490
				(2.071–3.950)	(1.392–4.453)
	Supportive care	139.0	43	<0.0001;	<0.0001;
				4.443	5.504
				(2.558–7.718)	(3.126–9.689)

Table 4: Independent prognostic factors. Median survival and significance displayed for each group across whole cohort. The median age of the entire cohort was 70.91. G12D, G12V and G12R were compared with all other *KRAS* and *KRAS* wild-types. Univariate analysis performed using log-rank test; multivariate analysis performed using Cox proportional hazards model. Multivariate analysis for lymph node status, tumour location, neoadjuvant therapy, margin status and adjuvant therapy performed with the surgery group. WT, wild-type; HR, hazard ratio; 95% CI, 95% confidence interval.

**Table 5 cells-11-03175-t005:** Effect of surgical factors on survival in surgical patients.

		Median Survival (Days)	Number of Patients	Univariate Analysis (*p*-Value; HR 95% CI)	Multivariate Analysis (*p*-Value; HR 95% CI)
Lymph node status	Negative	Undefined	13	Reference	Reference
	Positive	729.0	37	0.0176;	0.543;
				3.235	1.621
				(1.572–6.656)	(0.342–7.681)
Tumour location	Head	720.0	40	Reference	Reference
	Distal	Undefined	13	0.0118;	0.073;
				0.2878	0.290
				(0.1414–0.5860)	(0.075–1.124)
Neoadjuvant therapy	No	810.0	27	Reference	Reference
	Yes	1028	24	0.2943;	0.361;
				0.6083	2.075
				(0.2726–1.357)	(0.433–9.948)
Surgical margins	Clear	1016	10	Reference	Reference
	Positive	729.0	43	0.2018;	0.156;
				1.532	1.849
				(0.7855–2.988)	(0.791–4.323)
Adjuvant therapy	No	636.5	12	Reference	Reference
	Yes	1016	39	0.3166;	0.108;
				0.7894	0.360
				(0.3225–1.932)	(0.104–1.249)

Table 5: Independent prognostic factors. Median survival and significance displayed for each group with extra surgical factors in patients who received surgery. Univariate analysis performed using log-rank test; multivariate analysis performed using Cox proportional hazards model. Multivariate analysis for lymph node status, tumour location, neoadjuvant therapy, margin status and adjuvant therapy performed with the surgery group. HR, hazard ratio; 95% CI, 95% confidence interval.

**Table 6 cells-11-03175-t006:** Survival within stages and treatment groups.

			Number of Patients	Median Survival (Days)	Univariate Analysis (*p*-Value; HR 95% CI)	Multivariate Analysis (*p*-Value; HR 95% CI)
Resectable	*KRAS*	WT	8	1143	Reference	Reference
		Mutant	55	494.0	0.1296;	0.306;
					1.994	1.647
					(1.008–4.239)	(0.633–4.285)
	G12D	No	35	810.0	Reference	Reference
		Yes	28	356.0	0.0168;	0.019;
					2.053	1.991
					(1.139–3.702)	(1.121–3.537)
Borderline Resectable	*KRAS*	WT	0	-	-	-
		Mutant	23	296.0	-	-
	G12D	No	14	286.5	Reference	Reference
		Yes	9	358.0	0.3180;	0.113;
					0.6083	0.377
					(0.2347–1.576)	(0.113–1.259)
Locally Advanced	*KRAS*	WT	12	248.0	Reference	Reference
		Mutant	55	316.0	0.5046;	0.217;
					0.7941	0.634
					(0.3788–1.665)	(0.307–1.308)
	G12D	No	22	326.0	Reference	Reference
		Yes	45	316.0	0.2745;	0.216;
					1.351	1.441
					(0.7502–2.432)	(0.808–2.568)
Metastatic	*KRAS*	WT	9	196.0	Reference	Reference
		Mutant	69	189.0	0.5584;	0.874;
					1.228	1.068
					(0.5995–2.579)	(0.475–2.401)
	G12D	No	44	196.0	Reference	Reference
		Yes	34	190.0	0.6529;	0.815;
					1.114	0.941
					(0.6872–1.805)	(0.562–1.574)
Surgery	*KRAS*	WT	9	1143	Reference	Reference
		Mutant	52	729.0	0.2470;	0.438;
					1.717	1.459
					(0.7920–3.721)	(0.562–3.787)
	G12D	No	32	1084	Reference	Reference
		Yes	29	462.0	0.0221;	0.254;
					2.000	1.471
					(1.079–3.707)	(0.758–2.855)
Chemotherapy and/or Radiotherapy	*KRAS*	WT	14	225.0	Reference	Reference
		Mutant	111	290.0	0.7631;	0.782;
					1.089	1.090
					(0.6208–1.911)	(0.594–1.998)
	G12D	No	79	260.0	Reference	Reference
		Yes	46	310.5	0.4619;	0.364;
					1.156	1.207
					(0.7723–1.730)	(0.804–1.813)
Supportive Care	*KRAS*	WT	6	97.00	Reference	Reference
		Mutant	37	140.0	0.3007;	0.283;
					0.6427	0.591
					(0.2310–1.788)	(0.226–1.544)
	G12D	No	27	138.0	Reference	Reference
		Yes	17	135.0	0.6952;	0.696;
					1.131	1.136
					(0.5980–2.138)	(0.599–2.155)

Table 6: Survival within stages and treatment groups. Median survival and significance displayed for each group. Univariate analysis performed using log-rank test; multivariate analysis performed using Cox proportional hazards model. WT, wild-type; HR, hazard ratio; 95% CI, 95% confidence interval.

## Data Availability

The data presented in this study are available on request from the corresponding author. The data are not publicly available in accordance with consent provided by participants on the use of confidential data.

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
