# Peer review of "KRAS G12D Mutation Subtype in Pancreatic Ductal Adenocarcinoma: Does It Influence Prognosis or Stage of Disease at Presentation?"

_cells, 2022, doi:10.3390/cells11193175_

Round 1
Reviewer 1 Report
This study by Shen et al. conducted KRAS mutation analysis on FNA samples and resection specimens in pancreatic ductal adenocarcinoma. The authors have retrospectively evaluated 231 patients with cytological material and pancreatic resections for different types of KRAS mutations and claim that KRAS G12D is a prognostic biomarker in PDAC patients.
The study design is in general valuable, because FNA really is a good method and the concept of preoperative testing could enable a more stratified and personalized treatment. I see also a potential in the diagnostic value of KRAS testing (i.e. compared to finding of GNAS) not only in FNA material but also liquid biopsies.
Introduction: Acceptable summary of previous findings. Why not (instead of partially confusing paragraph on prognostic findings) also mention prediction value of G12C or even the diagnostic value of KRAS (compared GNAS mutations)?
line 58: that mutations of KRAS, CDKN2A, P53 and SMAD4 are associated with particular phenotypes has to be explained further. What is meant with phenotype? For histological phenotypes, there is no stringent genotype-phenotype correlation. In the reference that is given, I could not find such an association.
There is a longer passage about prvious finding concering prognostic value of G12 mutations. A real value to the reader would be to discriminate between real prognostic value (in multivariate analysis) and a trend (univariate analysis).
Methods section: To use NCCN clinical staging might be acceptable, though TNM of UICC seems much more common. IF used, please explain further: What are criteria for resectable vs. non-resectable? which version of the staging was used (Date?)
line 159: R-status is missing. What about tumor regression in neoadjuvant treated patients?
Results section: The tables look not formatted at all, its seems they are just copied out of some statistics program. Heading is missing. It is hard to keep the line and columns while reading. It is almost impossible to detect the bullet points.
Major concern: from an epidemiological point of view, one cannot claim the prognostic value of a biomarker if its only significant in univariate analysis. It must be independent. The maximum to say is, that there is a trend. Anyway, there is roughly only about 50 patients in the resection group.
Therefore Discussion line 350-352 must be rewritten, the results do not justify such a consideration. I think it is a statistical misconception that results could "get" significant if there were more patients.
Author Response
This study by Shen et al. conducted KRAS mutation analysis on FNA samples and resection specimens in pancreatic ductal adenocarcinoma. The authors have retrospectively evaluated 231 patients with cytological material and pancreatic resections for different types of KRAS mutations and claim that KRAS G12D is a prognostic biomarker in PDAC patients.
The study design is in general valuable, because FNA really is a good method and the concept of preoperative testing could enable a more stratified and personalized treatment. I see also a potential in the diagnostic value of KRAS testing (i.e. compared to finding of GNAS) not only in FNA material but also liquid biopsies.
Introduction: Acceptable summary of previous findings. Why not (instead of partially confusing paragraph on prognostic findings) also mention prediction value of G12C or even the diagnostic value of KRAS (compared GNAS mutations)?
Response: Thank you for this comment. We have revised the paragraph regarding the prognostic implications of KRAS mutations to improve clarity (lines 81-93). Unfortunately, we feel that space will not allow a treatment of other mutations such as GNAS within the introduction. Furthermore, GNAS mutations are not among the most common mutations in PDAC, and GNAS mutation frequency was not collected as part of the routine care for the cohort of patients included in this study. GNAS is not as well characterised as a biomarker in PDAC although it is much more common in IPMN and while it may be of potential interest as a diagnostic biomarker, this is beyond the scope of our study. The diagnostic utility of KRAS analysis has been extensively explored elsewhere, and is discussed in lines 73-79. With regards to the predictive value of KRAS G12C, we agree that this finding is of clear therapeutic significance and is discussed in our discussion (lines 361-365). Due to the relative rarity of the G12C subtype in PDAC, our study was not adequately powered to detect any prognostic significance of this or any other less common KRAS mutation subtypes, although this would be of interest in larger studies and meta-analyses.
line 58: that mutations of KRAS, CDKN2A, P53 and SMAD4 are associated with particular phenotypes has to be explained further. What is meant with phenotype? For histological phenotypes, there is no stringent genotype-phenotype correlation. In the reference that is given, I could not find such an association.
Response: Thank you for this comment. This statement was amended to replace “phenotype” with “oncogenic cellular processes” and the reference has been updated to be more relevant (lines 58-59).
There is a longer passage about prvious finding concering prognostic value of G12 mutations. A real value to the reader would be to discriminate between real prognostic value (in multivariate analysis) and a trend (univariate analysis).
Response: As described above, we have amended this section for clarity (lines 81-93).
Methods section: To use NCCN clinical staging might be acceptable, though TNM of UICC seems much more common. IF used, please explain further: What are criteria for resectable vs. non-resectable? which version of the staging was used (Date?)
Response: Thank you for your comment. We have used the clinical staging NCCN (2016) rather than TNM given that the latter can only accurately assessed in patients who underwent resection which is a minority (~20%). We have amended the methods to include the version used (lines 126-127) and justified this approach in the discussion (lines 330-335). We have also provided additional references to the most recent 2022 guidelines in the discussion.
line 159: R-status is missing. What about tumor regression in neoadjuvant treated patients?
Response: Resection margin clearance has now been clarified in the methods (line 155). Tumour regression was not consistently objectively graded in the neoadjuvant treated patients, and thus we were unable to include this as a comparison within the limits of this retrospective study which relied on targeted review of standard of care medical records. We agree this is an interesting phenomenon to explore in broader studies.
Results section: The tables look not formatted at all, its seems they are just copied out of some statistics program. Heading is missing. It is hard to keep the line and columns while reading. It is almost impossible to detect the bullet points.
Response: Thank you for this comment. The tables have been re-formatted now.
Major concern: from an epidemiological point of view, one cannot claim the prognostic value of a biomarker if its only significant in univariate analysis. It must be independent. The maximum to say is, that there is a trend. Anyway, there is roughly only about 50 patients in the resection group.
Therefore Discussion line 350-352 must be rewritten, the results do not justify such a consideration. I think it is a statistical misconception that results could "get" significant if there were more patients.
Response: Thank you for your comments. We have incorporated your feedback and this sentence now reads “There was also a suggestion (on univariate analysis) of reduced survival in patients with KRAS G12D mutation who underwent resection, however after adjusting for potential confounders this was no longer statistically significant (on multivariate analysis). A larger sample size may help clarify this further.” (lines 300-303)
Reviewer 2 Report
In this manuscript by Shen et al., the authors present the finding that patients with KRAS G12D mutated, resectable PDAC have lower survival than patients without G12D mutation.
They conduct a study on the KRAS mutation study in specimen of patients after surgical resection or fine needle aspiration.
The manuscript is well written and occurs to be well-executed and it raises several potentially interesting points of discussion, e.g., different KRAS variants and their potential influence on patient outcome or the use of fine needle aspiration vs. complete resection for resectable and non-resectable disease for diagnostic purposes. However, the manuscript as such is lacking overall depth and a clear line. Experimental data are limited to the KRAS mutation status, where data on the histology of the different specimen (Ki67 staining) would have been easy to obtain. Moreover, the significant association of KRAS G12D with lower survival is not very convincing, since G12D patients were compared non G12D and not KRAS WT patients. Therefore, major revision would be required addressing these issues.
Author Response
In this manuscript by Shen et al., the authors present the finding that patients with KRAS G12D mutated, resectable PDAC have lower survival than patients without G12D mutation.
They conduct a study on the KRAS mutation study in specimen of patients after surgical resection or fine needle aspiration.
The manuscript is well written and occurs to be well-executed and it raises several potentially interesting points of discussion, e.g., different KRAS variants and their potential influence on patient outcome or the use of fine needle aspiration vs. complete resection for resectable and non-resectable disease for diagnostic purposes. However, the manuscript as such is lacking overall depth and a clear line. Experimental data are limited to the KRAS mutation status, where data on the histology of the different specimen (Ki67 staining) would have been easy to obtain. Moreover, the significant association of KRAS G12D with lower survival is not very convincing, since G12D patients were compared non G12D and not KRAS WT patients. Therefore, major revision would be required addressing these issues.
Response: Thank you for your review. We have now undertaken a major revision to clarify the thought process behind this manuscript. We feel that we have demonstrated the following:
- That KRAS G12D is associated with a reduction in survival in patients with resectable disease
- That KRAS mutation determination can be accurately assessed using EUS-FNA material
- That 1 and 2 allows us to incorporate potentially important prognostic (KRAS G12D) and predictive (KRAS G12C) information into the clinical management of patients with pancreatic cancer.
Given the contradictory results in the literature around the impact of KRAS G12D on survival we feel that our data adds weight to the argument that it is an indicator of poor prognosis in early-stage disease. A meta-analysis, brining all this data together (along with our contribution) would help to clarify this question.
Furthermore, this study was conducted as a retrospective review of medical records and KRAS mutation status was the only biomarker that was consistently tested for throughout the study period, largely as part of an active clinical trial in the local institution. Neither KRAS mutation testing or Ki67 are routinely collected as part of standard of care in the local institutions included in this study, and as such we do not have access to this information (lines 356-359). With regards to our survival analysis, G12D patients were compared to all non G12D patients which included all other KRAS mutations and KRAS WT. We have clarified this in methods and results sections. We recognize the limitations of our sample size, particularly when comparing less common molecular subtypes, and reference this in our discussion section.
Reviewer 3 Report
The manuscript by Shen Dr al describes a retrospective evaluation of KRAS mutation in PDAC and a variety of predictive or prognostic correlates, including stage of disease, overall survival and response to treatment. Overall, the authors found that the G12D mutation, still identified as the most common mutation in PDAC, predicted poorer survival in patients with respectable PDAC versus those with either other KRAS mutations or with WT KRAS. Overall the paper muddies the utility of KRAS mutation subtype in analyses of patient prognosis or treatment options, and while it identifies a correlation in a particular sub population that may be able to be acted upon with more aggressive therapy, there is minor value added to the overall scientific community with these findings. There are a few suggestion that may strengthen the manuscript, which, if addressed, would increase the enthusiasm for this work.
Suggestions:
The largest suggestion is to broaden the scope of analyses of KRAS mutant type to at least the three most common mutations, being G12D, V, and R. While the authors discuss the sample sizes being too small to evaluate each mutant type, these three common ones deserve more attention.
There is also a variation in comparators - sometimes the authors compare all mutant KRAS to WT, and sometimes it is G12D versus others. It would improve the authors communication with the readers to increase consistency.
Also, it would be valuable for each table entry to include the number of patients which fall into each category - particularly for Table 3.
If possible, the authors would add value to the manuscript to discuss what treatments were employed in the treatment groups and compare efficacy by KRAS subtype (if the numbers allow for this analysis). Time to recurrence would also be valuable to consider, rather than just median survival.
Minor points:
The choice of age differentiation at 70.91 years isn’t clear - please explain
What are the difference in TMN versus NCCN staging that lends more value to NCCN? Please expand that discussion.
The authors all use in lines 348-350 that G12D analysis in borderline resettable patients may help guide the decision to resect, but the data presented showed a loss of G12D prognostic value in this group, so their data doesn’t support this conclusion.
Author Response
The manuscript by Shen Dr al describes a retrospective evaluation of KRAS mutation in PDAC and a variety of predictive or prognostic correlates, including stage of disease, overall survival and response to treatment. Overall, the authors found that the G12D mutation, still identified as the most common mutation in PDAC, predicted poorer survival in patients with respectable PDAC versus those with either other KRAS mutations or with WT KRAS. Overall the paper muddies the utility of KRAS mutation subtype in analyses of patient prognosis or treatment options, and while it identifies a correlation in a particular sub population that may be able to be acted upon with more aggressive therapy, there is minor value added to the overall scientific community with these findings. There are a few suggestion that may strengthen the manuscript, which, if addressed, would increase the enthusiasm for this work.
Suggestions:
The largest suggestion is to broaden the scope of analyses of KRAS mutant type to at least the three most common mutations, being G12D, V, and R. While the authors discuss the sample sizes being too small to evaluate each mutant type, these three common ones deserve more attention.
Response: Thank you for this feedback. We have now included assessment of the impact of G12V and G12R on survival. Neither was associated with an obvious impact on survival when considering the whole population (all stages of disease) (lines 228-229). With the small samples sizes and the median survival with and without each of these subtypes being so similar, we did not feel that it was worthwhile to assess for survival within clinical stages. The text has been amended to reflect these results.
There is also a variation in comparators - sometimes the authors compare all mutant KRAS to WT, and sometimes it is G12D versus others. It would improve the authors communication with the readers to increase consistency.
Response: Thank you for this comment. Throughout the manuscript, the total KRAS mutant population was compared with KRAS WT. The G12D populations was compared to non G12D population which consists of both those patients with alternative mutant KRAS subtypes and those with WT disease. This has now been clarified in both the methods and results.
Also, it would be valuable for each table entry to include the number of patients which fall into each category - particularly for Table 3.
Response: Thank you for this feedback – we have updated tables 4 (formerly 3), 5 and 6 to include this information.
If possible, the authors would add value to the manuscript to discuss what treatments were employed in the treatment groups and compare efficacy by KRAS subtype (if the numbers allow for this analysis). Time to recurrence would also be valuable to consider, rather than just median survival.
Response: Thank you for your comment. We have amended the limitations section of the discussion to acknowledge and explain these shortcomings. “Finally, the exact details of the chemotherapy received were not reported as many patients had been referred from other centers for diagnostic biopsy and returned to the referring center for chemotherapy. Furthermore, the chemotherapy regimens used were too heterogenous and, given the sample size, were unlikely to have provided statistically meaningful results when examined individually. Time to recurrence was also not reported as it is difficult to collect in retrospective studies especially as many patients returned to their referring centers for ongoing treatment.” (lines 384-390)
Minor points:
The choice of age differentiation at 70.91 years isn’t clear - please explain
Response: 70.91 was the median age of the entire cohort. This has now been indicated in the table explanation for table 4.
What are the difference in TMN versus NCCN staging that lends more value to NCCN? Please expand that discussion.
Response: Thank you for this comment. We clarified why NCCN is more appropriate than the TNM in our population. TNM can only speak to the 20% of patients who undergo resection where NCCN is a clinical staging system that can be applied to all patients and is widely used in clinical practice (lines 330-335).
The authors all use in lines 348-350 that G12D analysis in borderline resettable patients may help guide the decision to resect, but the data presented showed a loss of G12D prognostic value in this group, so their data doesn’t support this conclusion.
Response: Thank you for this comment. However, we do note that the presence of KRAS G12D is associated with worse survival on univariate analysis in the resected cohort. When analysed with a multivariate analysis (incorporating lymph node status, margin and adjuvant therapy) it was no longer statistically significant. However, none of these factors can be known for certain prior to resection and therefore it may be more relevant in this situation to accept the results of the univariate analysis? Furthermore, some of the patients who went on to surgical resection were initially in the borderline resectable group which further complicates the interpretation of our results. Therefore, we feel that it is reasonable to use the data we have presented which clearly show that KRAS G12D is a poor prognostic factor in clearly operable pancreatic cancer to potentially inform decision regarding surgery in clearly operable patients.
Round 2
Reviewer 1 Report
Thanks for you responses. The authors have answered all questions and have re-written passages.
My only concern (still) is to say that the survival is significantly shorter (abstract, conclusions).
There is a potential mis-spelling in the Affiliations: might be "Innate" immunitiy.
line 96: word is missing to complete sentence, "correlation"?
line 146: Which anatomic criteria? Involvement of V. mesenterica? Liver metastasis? Please explain at least briefly.
lin 370-375: Is there a chance that simply the complications/morbidity of the whipple surgery (or pancreatectomy) could explain the shorter survival of resected patients???
Author Response
Thanks for you responses. The authors have answered all questions and have re-written passages.
My only concern (still) is to say that the survival is significantly shorter (abstract, conclusions).
Thank you very much for all of your comments over the course of the submission process, they have been extremely helpful in improving the quality of the paper.
We acknowledge there may be confusion and have therefore removed some of the “significant” from the abstract and conclusions. However, we did find that resectable G12D patients had shorter survival on multivariate analysis, and therefore this is a statistically significant finding. We agree that more data is required to definitively make this conclusion, hence we plan to conduct a meta-analysis including this study if successful.
There is a potential mis-spelling in the Affiliations: might be "Innate" immunitiy.
Thank you for pointing this out, we have now fixed the spelling, it should indeed read Innate Immunity!
line 96: word is missing to complete sentence, "correlation"?
Thank you for this suggestion, with regards to this sentence about liquid biopsies, we have clarified the meaning to make the sentence more complete.
line 146: Which anatomic criteria? Involvement of V. mesenterica? Liver metastasis? Please explain at least briefly.
Thank you for this comment. The NCCN anatomical criteria have now been briefly explained in lines 126-129, where we give an example of vessel involvement to stratify between clinical stages. The full criteria is quite extensive and detailed, and thus we have included reference to the full guidelines.
lin 370-375: Is there a chance that simply the complications/morbidity of the whipple surgery (or pancreatectomy) could explain the shorter survival of resected patients???
Thank you for this insight. Regarding lines 370-375 in the original resubmission, we had initially calculated post surgery mortality of G12D vs non G12D, and found no difference. Furthermore, given that resectability and other surgical factors were not different between the two groups, we acknowledge that it is unlikely that complications/morbidity of surgery explain the survival difference. However, we agree that this may be interesting to investigate in the future, given a larger sample size and more complete follow-up.
Reviewer 3 Report
All of my previous concerns have been addressed.
Author Response
Thank you very much for your comments, they have been extremely helpful in improving the quality of the paper.